# Genetic Dissection of Temperament Personality Traits in Italian Isolates

**DOI:** 10.3390/genes13010004

**Published:** 2021-12-21

**Authors:** Maria Pina Concas, Alessandra Minelli, Susanna Aere, Anna Morgan, Paola Tesolin, Paolo Gasparini, Massimo Gennarelli, Giorgia Girotto

**Affiliations:** 1Institute for Maternal and Child Health-IRCCS “Burlo Garofolo”, 34127 Trieste, Italy; anna.morgan@burlo.trieste.it (A.M.); paolo.gasparini@burlo.trieste.it (P.G.); giorgia.girotto@burlo.trieste.it (G.G.); 2Department of Molecular and Translational Medicine, University of Brescia, 25123 Brescia, Italy; alessandra.minelli@unibs.it (A.M.); massimo.gennarelli@unibs.it (M.G.); 3Genetics Unit, IRCCS Istituto Centro S. Giovanni di Dio Fatebenefratelli, 25125 Brescia, Italy; 4Department of Medicine, Surgery and Health Sciences, University of Trieste, 34139 Trieste, Italy; SUSANNA.AERE@studenti.units.it (S.A.); paola.tesolin@burlo.trieste.it (P.T.)

**Keywords:** temperament scales, genome-wide association studies, *MAGI2*, *CALCB*, *BTBD3*, *PARK2*

## Abstract

Human personality (i.e., temperament and character) is a complex trait related to mental health, influenced by genetic and environmental factors. Despite the efforts performed during the past decades, its genetic background is only just beginning to be identified. With the aim of dissecting the genetic basis of temperament, we performed a Genome-Wide Association Study (GWAS) on Cloninger’s Temperament and Character Inventory in 587 individuals belonging to different Italian genetic isolates. Data analysis led to the identification of four new genes associated with different temperament scales, such as Novelty Seeking (NS), Harm Avoidance (HA), and Reward Dependence (RD). In detail, we identified suggestive and significant associations between: *MAGI2* (highest *p*-value = 9.14 × 10^−8^), a gene already associated with schizophrenia and depressive disorder, and the NS–Extravagance scale; *CALCB* (highest *p*-value = 4.34 × 10^−6^), a gene likely involved in the behavioral evolution from wild wolf to domestic dog, and the NS–Disorderliness scale; *BTBD3* (highest *p*-value = 2.152 × 10^−8^), a gene already linked to obsessive–compulsive disorder, and the HA–Fatigability scale; *PRKN* (highest *p*-value = 8.27 × 10^−9^), a gene described for early onset Parkinson’s disease, and the RD scale. Our work provides new relevant insights into the genetics of temperament, helping to elucidate the molecular basis of psychiatric disorders.

## 1. Introduction

Human personality is a complex trait resulting from the innate predisposition to respond to external stimuli and interaction with environmental factors [1]. Previous studies on families, twins, and adopted children demonstrated that personality is strongly biologically determined, showing that its heritability ranges between 30% and 60% [2,3]. However, to date, genome-wide association studies (GWAS) allowed only a limited portion of the heritable component to be highlighted, suggesting that the understanding of the link between genetic variants and personality remains challenging [4].

Methods that allow us to define and classify personality are necessary to understand its underlying biology. For this specific purpose, Robert Cloninger developed the Temperament and Character Inventory (TCI) by analyzing both individuals with personality disorders and the general population [5]. Today, the TCI is one of the most commonly used models to assess personality. In particular, the answers to a 240-item questionnaire allow assessment of two domains of personality: temperament and character. These two dimensions of personality seem to rely on different genetic and neuronal pathways: character networks are linked to intentional and meta-cognitive processes (self-reflection, goal setting, empathy, episodic learning), while temperament is associated with basic emotions (e.g., stress reactions such as fear, anger, disgust) [3]. Briefly, individuals′ scores at the TCI allow their sorting into four classes of temperaments and three classes of characters and the different subscales of both.

In this work, we focused on the genetic architecture of temperament, considering both its scales and subscales. In particular, we analyzed the scales of Novelty Seeking (NS), Harm Avoidance (HA), Reward Dependence (RD), and Persistence (P), and their relative subscales, as defined by Cloninger [2]. TCI scores and Temperaments’ traits have been already studied in the context of personality and psychiatric disorders. For example, low scores of NS are linked to autism spectrum disorder [6], while high HA scores have been registered in patients affected by Parkinson′s disease and obsessive–compulsive disorders [7,8]. These results suggest that a specific temperament can be viewed as a susceptibility factor for psychiatric morbidity [6,9] and thus that genes′ variants potentially affecting temperament traits could also exert a role in related disorders.

Based on these considerations, we performed a GWAS to define the genetic base of temperament by employing the data available for a large cohort of subjects coming from Italian genetic isolates. To our knowledge, this is one of the few GWAS exploring the genetic asset of temperament that takes advantage of isolated populations, which are known for being characterized by a low genetic variability and an elevated environmental homogeneity.

## 2. Materials and Methods

### 2.1. Description of the Sample

This study was conducted on a cohort of 587 individuals belonging to six villages of the Friuli Venezia Giulia (FVG) region, in the North of Italy. The six villages are: Clauzetto, Erto e Casso, Illegio, Resia, Sauris, and San Martino del Carso. FVG is part of the Italian Network of Genetic Isolates (INGI) Project, a collaboration between research institutions in Italy aimed at detecting the molecular bases of complex traits by investigating genetically isolated Italian populations.

A detailed description of these populations has been previously reported [10,11,12]. The ethical committees of the IRCCS Burlo Garofolo approved the study, and all participants signed written informed consent.

### 2.2. Personality Assessment

All subjects completed the TCI to assess the four dimensions of temperament (Novelty Seeking, Harm Avoidance, Reward Dependence, and Persistence), as shown in Figure 1.

All subjects completed the Cloninger′s TCI answering true or false to 240 items with the following indications: “Answer true or false depending on whether the statement describes your attitude or not, read each sentence carefully but decide quickly, answer each question even if you are not completely sure”. True or false answers can assign one point to the subscales. The sum of the subscale scores gives the overall value of the scale. Higher scores indicate greater relevance of the scale. Particularly in the temperament scales, the items are distributed in this way:-NS is measured using 40 items; of these, 11 for Excitability, 10 for Impulsiveness, 10 for Extravagance, and nine for Disorderliness;-HA is measured using 35 items; of these, 11 for Pessimism (anticipatory worry), seven for Fearfulness (fear of uncertainty), eight for Shyness, and nine for Fatigability;-RD is measured using 24 items; of these, 10 for Sentimentality, 8 for Attachment, 6 for Dependence (on approval of others);-P is measured using eight items.

All the scales range between 0 and 100. The score for each of 15 scales was calculated and studied in the genetic analysis as quantitative traits.

### 2.3. Anxiety and Depression Evaluation

The participants received a structured diagnostic interview using the Composite International Diagnostic Interview (CIDI) in order to assess current and lifetime diagnoses according to the *Diagnostic and Statistical Manual of Mental Disorders—Fourth edition—Text Revision* (DSM-IV-TR).

### 2.4. Inter-Correlations between the Temperament Scales and Associated Variables

First, scales and subscales’ mean and their distributions were examined, and Pearson′s correlation coefficients between scales were calculated. In order to assess variables influencing the temperament scales, linear regression models were applied. Age and sex were evaluated for each scale, while education level was tested for NS and anxiety or depression status for HA scales. All the analyses were performed in R v.4.0.0. (www.r-project.org, accessed on 10 May 2021).

### 2.5. Genotyping and Imputation

All individuals included in this study have been genotyped with Illumina 370 k/700 k/MEGA (~1,800,000 SNPs) high-density SNP array (Illumina Inc., San Diego, CA, USA). Genotypes were called with Illumina GenomeStudio. Each batch was processed according to standard quality control (QC) procedures with the following criteria for inclusion: sample call-rate ≥ 0.95, gender check, SNP call rate ≥ 0.95, Hardy–Weinberg Equilibrium *p*-value > 1 × 10^−6^, and minor allele frequency (MAF) ≥ 0.01. The number of SNPs used for imputation was 330,151 for 370K, 633,354 for 700K, and 1,233,355 for MEGA. Genotype imputation was conducted in the three batches separately, using IMPUTE2 [13] considering as reference a customer panel generated by the 1000 Genomes phase 3 and whole genome sequences of INGI samples [12]. After imputation all batches were merged and info score and allele frequencies were calculated using QCTOOL software v2 (https://www.well.ox.ac.uk/~gav/qctool_v2/, accessed on 12 February 2020). A total number of about 89 million variants were obtained at this step. After imputation, SNPs with MAF < 0.01 and Info Score < 0.4 were discarded from the statistical analyses. The Info Score is an information metric that IMPUTE2 reports, and it is used to remove poorly imputed SNPs from the association testing analysis. Indeed, this metric assumes values between zero and one, where values near one indicate that a SNP has been imputed with high certainty. Generally, a SNP with an Info Score greater than 0.4 was considered as an acceptable well-imputed variant.

A total number of 10,889,024 SNPs were analyzed.

### 2.6. Genome-Wide Association Study

The GWAS for all the 15 scales were conducted using mixed linear models with the temperament scales as dependent variables and the allele dosages as the independent variable. The genomic kinship matrix, based on all available genotyped SNPs, was used as the random effect to take into account the relatedness. Association analyses were conducted using the GRAMMAR-γ method [14], as implemented in the GenABEL R package [15]. GenABEL was used to eliminate the effect of relatedness from the traits and MixABEL R package [15] was used for the actual association of the imputed SNPs. All models were adjusted by sex and age. Considering the effect of educational level on NS scales, the analyses for these traits were also adjusted for years of schooling. Anxiety and depression were used as covariates for HA traits.

Genome-wide significance was set to 5 × 10^−8^; as the traits we analyzed were not independent [16,17], we did not conduct any further correction of the significance threshold for multiple testing. An association was considered suggestive if it presented a *p*-value < 1 × 10^−5^. SNPs were annotated with the Variant Effect Predictor tool (VEP, https://www.ensembl.org/info/docs/tools/vep/index.html [18], accessed on 7 September 2021), to determine their distance from the closest genes in a range of 250 kb and to obtain their functional characteristics (i.e., whether they were contained within an intronic, exonic, or intergenic region) and genes biotype. Long non-coding RNA (LINC) genes or genes with unknown function identified with LOC and FAM symbols or pseudogenes were excluded, and we focused on the protein coding genes. We applied the following approach to prioritize the genes: 1) we excluded the genes with only one SNP with *p*-value < 1 × 10^−5^ in around 250 kb; 2) between the remaining genes, we selected all those with SNPs showing *p*-value < 1 × 10^−7^; 3) if no SNPs near/within gene had *p*-value < 1 × 10^−7^, we considered genes with more than 50 SNPs showing *p*-value < 1 × 10^−5^. All data are aligned to the Human genome reference build 37 (GRCh37).

### 2.7. Expression Level Analysis

For the top SNP of each identified gene, the association with the expression in brain tissues of the gene was checked using GTEx (https://www.gtexportal.org/home/, [19], accessed on 15 September 2021). GTEx eQTL browser is a resource that reports the results of a research project for determining the association between genetic variation and high-throughput molecular-level expression phenotypes. This information can help to better understand the biological relevance of results from GWAS. Details on the procedure used to obtain eQTLs were published in https://www.gtexportal.org/home/documentationPage (accessed on 15 September 2021). Briefly, RNA-seq data were analyzed and mapped using FastQTL [20], considering the covariates described at https://www.gtexportal.org/home/documentationPage (accessed on 15 September 2021). Nominal *p*-values were generated for each variant gene pair by testing the alternative hypothesis that the slope of a linear regression model between genotype and expression deviates from zero in a map window of 1Mb. βdistribution-adjusted *p*-values were used to calculate *q*-values [21] and a false discovery rate threshold of ≤ 0.05 was used to define genes with significant eQTL.

## 3. Results

A total of 587 individuals (56.2% women, age: range 18–82 years, mean 48.6, standard deviation 14.7) were included in the analyses. The mean years of schooling was 11.69 (standard deviation 4.2).

Appendix A provides a description of the investigated traits in terms of mean and standard deviation, while Appendix A shows the Pearson’s correlations between the scales. We found negative significant (*p*-value < 0.05) correlations between NS (Novelty Seeking) and HA (Harm Avoidance) scales, NS and P (Persistence) (except NS1 for which the correlation whit P was positive), HA and P, and HA and RD3 (Attachment). Positive significant correlations were found between NS and RD (Reward Dependence) (except NS2, for which the correlation was negative), RD and P, and HA and RD1 (Sentimentality). RD showed significant positive correlation between HA2 (Fearfulness) and negative between HA3 (Shyness).

By means of regression models, reported in Appendix A, we observed that, compared to females, males showed significantly (*p*-value < 0.05) lower scores for RD (Reward Dependence), RD1 (Sentimentality), HA (Harm Avoidance), HA2 (Fearfulness), HA3 (Shyness), HA4 (Fatigability), and NS3 (Extravagance). With increasing age, we observed higher significant scores for RD1 (Sentimentality), HA2 (Fearfulness), HA4 (Fatigability), and lower scores for RD3 (Attachment), RD4 (Dependence), NS (Novelty Seeking), NS1 (Excitability), NS3 (Extravagance), and NS4 (Disorderliness). Individuals with a higher education level showed higher significant scores for NS and NS1 scales. Finally, individuals with anxiety or depression showed higher significant scores for all the HA scales. Because of the association with education level (available for 482 individuals), this variable was included as a covariate in the GWAS of NS scales. In addition, for the traits of the HA scale the information about anxiety and major depression disease was added in the models. The number of individuals for which this information is available was 528 and 191 were affected by anxiety or depression.

Regarding the genetic analysis, the effect of population stratification on the GWAS was negligible, as confirmed by the values of the genomic inflation factor (lambda range: 0.97–1.03). QQ and Manhattan plots are shown in Appendix A. Appendix A shows all the associated SNPs at *p*-value < 1 × 10^−5^, while in Appendix A their VEP annotation is available. In particular, we detected seven SNPs with genome-wide significant *p*-value (< 5 × 10^−8^), but only five were located 250 kb near protein coding genes. Two SNPs, found for RD scale phenotype, were located within *PARK2* gene (rs10455897, *p*-value 8.27 × 10^−9^, and rs6917337, *p*-value 4.91 × 10^−8^), two identified for HA4 trait were located upstream of the *BTBD3* gene (rs4814036, *p*-value 2.15 × 10^−8^, and rs8123788, *p*-value 3.80 × 10^−8^), while one (rs115928744, *p*-value 2.99 × 10^−8^) was found within the *DPYSL3* gene. In addition, we identified eight other SNPs with *p*-value < 1 × 10^−7^, seven of which were located near protein coding genes: five of them were located upstream *BTBD3* (HA4 trait) and one was within the *PARK2* gene (RD trait). The last one, found in the NS3 phenotype, was detected within the *MAGI2* gene (rs13242288, *p*-value 9.14 × 10^−8^). Moreover, as displayed in Appendix A, the signal in *BTBD3* was supported by a total of 72 SNPs with *p*-value < 1 × 10^−5^, the hit in *PARK2* by 27 SNPs and the one related to *MAGI2* by 24 SNPs. Instead, only one SNP was detected within *DPYSL3* gene and thus it was excluded from our prioritized list. Among suggestive signals, 81 SNPS, identified in the NS4 phenotype, were located 250 kb near the *CALCB* gene (top SNP rs10832317, *p*-value 4.56 × 10^−6^).

Thus, considering the procedure described in the Materials and Methods section, our analyses allowed the identification of four final genes, as displayed in Table 1.

Briefly, for the NS scales, our GWAS highlighted two extremely interesting protein coding genes: *MAGI2,* located within chromosome 7, and *CALCB* in chromosome 11.

*MAGI2* (encoding the membrane associated guanylate kinase, WW, and PDZ domain containing 2 protein) was identified in NS3 (Extravagance) scale analysis (Figure 2A). The same gene was also found to be associated with NS2 (Impulsiveness) through two intronic SNPs (top SNP rs12374970, *p*-value 3.45 × 10^−6^) (Appendix A).

The other gene identified in the NS GWAS and exactly in the NS4 subscale (Disorderliness) was *CALCB* (encoding the calcitonin-related polypeptide β protein) (Figure 2B).

Regarding the HA scales, we identified the gene *BTBD3* (BTB domain containing 3) in the GWAS of HA4 (Fatigability) (Figure 2C).

For RD (Reward Dependence) we found *PRKN* (encoding the parkin RBR E3 ubiquitin protein ligase protein, also named *PARK2*) gene, located in chromosome 6 (Figure 2D). In addition, the same gene signal was highlighted in RD3 (Attachment) GWAS by ten intronic SNPs (top SNP rs9346902, *p*-value 2.21 × 10^−7^) as displayed in Appendix A.

Regarding GTEx analysis, we found an association between the top SNP rs10832317 and the expression level of *CALCB* gene in brain–cerebellum. Precisely, the G allele is associated with a lower level of *CALCB* expression (NES (Normalized effect size) = −0.35, *p*-value 0.00012); the same G allele is associated in our analysis with higher scores for NS4 (Appendix A).

## 4. Discussion

In this work, we investigated the genetic basis of temperament by performing a GWAS on temperament scales and subscales in 587 individuals coming from Italian isolated populations.

Data analysis led to the identification of four new genes, *MAGI2*, *CALCB*, *BTBD3* and *PRKN*, associated with different personality traits such as NS, HA, and RD.

Regarding *MAGI2*, we identified a significant association between this gene and the NS3 subscale (Extravagance) (top SNP: rs13242288, *p*-value = 9.14 × 10^−8^). *MAGI2* encodes the membrane-associated guanylate kinase, WW and PDZ domain containing 2 protein, which seems to be specifically expressed in the brain, in particular in both excitatory and GABAergic synapses [22] and plays crucial roles in the assembly of signaling complexes [22]. Interestingly, previous literature data described an association of this gene with different diseases and/or psychiatric disorders. In particular, *MAGI2* has been associated with Alzheimer′s Disease (AD) [23] and it has been proposed as a candidate transcriptomic biomarker of AD [24]. Indeed, data from mouse models showed a significant age-dependent increase of *Magi2* expression in the blood, contextually with changes in the hippocampus [24]. Moreover, prior GWAS linked *MAGI2* to schizophrenia [25], evidence further supported by the detection of duplications and variations of this gene in schizophrenic patients [25,26]. Recent studies on personality in schizophrenia consistently found increased neuroticism, decreased extraversion, and decreased conscientiousness compared to normative levels or healthy controls, and these differences appear to persist through active and residual phases of the illness. Thus, these data might represent the starting point to understand the possible role of *MAGI2* in the Extravagance (NS3) temperament trait and its possible relationship with the associated diseases [5].

The second relevant finding concerns the *CALCB* gene, which encodes the calcitonin gene-related peptide 2, a protein widely expressed in the mammalian brain and involved in the regulation of vascularity and with a neuropeptide hormone activity. Here, we detected an interesting association between this gene and the Disorderliness (NS4) subscale (top SNP: rs10832317, *p*-value = 5.60 × 10^−6^), whose high scores are typical of subjects who tend to prefer activities without strict rules and do not like fixed routines [5].

An interesting study about dog domestication observed changes in the *CALCB* expression pattern in the hypothalamus of domestic dogs (*Canis familiaris*) compared to gray wolves (*Canis lupus*), two species with remarkable behavioral differences, despite the relatively recent divergence time [27]. In particular, the authors described different levels of *CALCB* hypothalamic expression, with higher levels in the domestic dog and lower in wolves, suggesting that changes in mRNA expression might affect behavioral character, contributing to the domestication process. Interestingly, the top SNP here identified (rs10832317) is an eQTL for *CALCB* in *Brain–Cerebellum*, with the G allele being associated with reduced gene expression. In this light, it is possible to speculate that a lower level of *CALCB* might be associated with a tendency to not stick to the rules, as it happens in individuals with NS4 high scores. Moreover, it has also been proposed that *CALCB* may play a role in the development of anxiety and depression [28], supporting the involvement of this gene in psychiatric disorders.

The third gene of interest is *BTBD3*, which encodes the BTB/POZ domain-containing protein 3. We identified a significant association between this gene and the HA4 subscale (Fatigability) (top SNP: rs4814036, *p*-value = 2.15 × 10^−8^). Individuals with HA4 high scores are usually asthenic, have less energy than most people and typically recover more slowly from minor illnesses or stress. On the opposite, individuals who obtain a low result on the Fatigability scale tend to be highly energetic and dynamic [5]. *BTBD3* seems to be involved in the dendritic field orientation during the development of the sensory cortex in mice [29], and previous GWAS described an association between *BTBD3* and obsessive–compulsive disorder (OCD) [30]. Interestingly, previous studies underlined the relationship between OCD and temperament traits, showing an important correlation with high HA scores [7]. In this light, our data further support the possible link between this gene, the HA scale, and ultimately psychiatric disorders.

Finally, we detected a significant association between *PRKN* and the RD (Reward Dependence) scale (top SNP: rs10455897, *p*-value = 8.27 × 10^−9^). Individuals with a high score in this scale tend to have an “addictions-prone” personality and to be easily influenced by others, with loss of objectivity and the need for emotional support and external approval [5]. The *PRKN* gene encodes the Parkin RBR E3 ubiquitin protein ligase, a protein involved in the pathway of protein ubiquitination. In particular it has been studied in the mitochondrial mechanisms of quality control and associated with the regulation of mitochondrial apoptosis and neurodegenerative diseases [31]. *PRKN* mutations are known to be involved with early onset familial and autosomal recessive juvenile Parkinson’s disease [32,33], likely due to the preferential degeneration of dopaminergic neurons in the substantia nigra pars compacta, resulting from the accumulation of damaged mitochondria [31]. Interestingly, dopamine plays a central role in the Reward system, which regulates the effects of decision-making and induces approach behavior. The implication of *PRKN* in the dopaminergic neurons is, therefore, very suggestive given the association with the RD scale here identified.

To summarize, we performed 15 GWAS, one for each subscale, considering as genome-wide significant the standard threshold of *p*-value < 5 × 10^−8^ instead of using the Bonferroni correction (*p*-value 3.3 × 10^−9^, i.e., 5 × 10^−8^/15) that would be overly stringent, all the traits being correlated with each other. On the other hand, although not independent, the temperament scales capture distinct constructs. Therefore, it is important to underline that signals within *PRKN* and *BTBD3* genes (*p*-value 8.27 × 10^−9^ and 2.15 × 10^−8^, respectively) can be considered as “genome-wide significant” but they have not reached the overcorrected threshold; thus a replication of these data would be performed in future works. 

## 5. Conclusions

During the past few decades, GWAS studies have made huge progress towards unraveling the genetic etiology of human personality, even though the exact genetic contribution has not yet been fully understood. The results of our study represent the first association of *MAGI2*, *CALCB*, *BTBD3,* and *PARK2* with different temperament scales, providing new insights into the genetics of this complex trait.

The availability of a cohort of individuals coming from isolated populations, with detailed phenotypic information, allowed us to improve the discovery power of GWAS even in the absence of a replica, which will be a future step of our work. In fact, isolated populations share an elevated environmental homogeneity and low genetic variability, leading to the identification of new polymorphisms with relevant impact on the complex trait of interest.

Our data look particularly promising, since all the genes identified here are expressed in the central nervous system, and their implication in several psychiatric disorders has already been suggested.

In this light, these data might help to draw the path from GWAS to biology and ultimately to the understanding of the genetic background of personality and psychiatric disorders.

## Figures and Tables

**Figure 1 genes-13-00004-f001:**
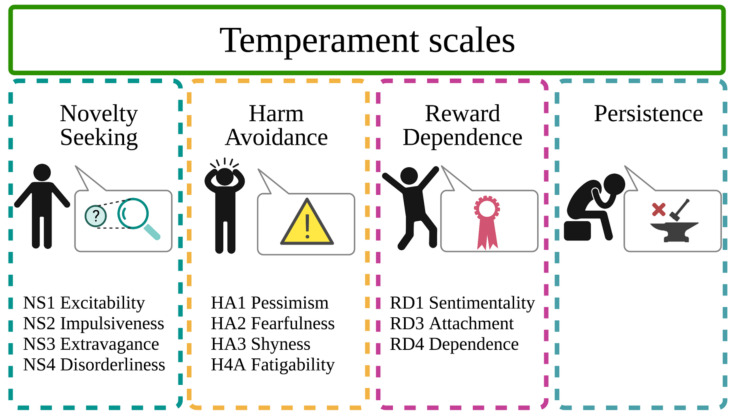
Temperament scales and subscales. The TCI defines four dimensions of temperament, namely Novelty Seeking, Harm Avoidance, Reward Dependence, and Persistence, each of them (except Persistence) further divided in different subscales.

**Figure 2 genes-13-00004-f002:**
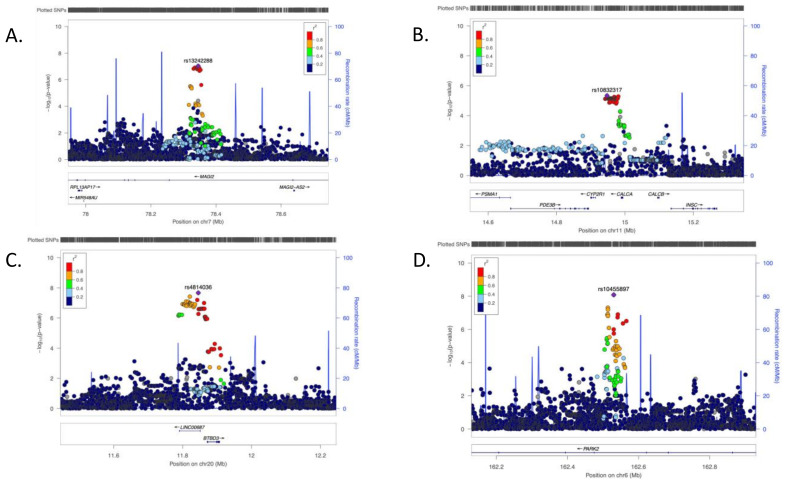
Regional association plot of the detected genes. It shows the regional association plot for each gene. (**A**) *MAGI2* found in the GWAS for NS3 (Extravagance); (**B**) *CALCB* for NS4 (Disorderliness); (**C**) *BTBD3* for HA4 (Fatigability); (**D**) *PARK2* for RD (Reward Dependence). Plots are produced in LocusZoom (http://locuszoom.org), accessed on 15 September 2021.

**Table 1 genes-13-00004-t001:** Genes highlighted by GWAS. The table shows the four genes identified by GWAS in the present study. The columns are: Phenotype: the phenotype in which the gene was identified; Nearest Gene (protein coding): the candidate gene closest to the top SNP identified; Chr: chromosome; Start position: start position of the nearest gene (build hg19); End position: end position of the nearest gene (build hg19); Top SNP: the most significant SNP; *p*-value: *p*-value of the top SNP; Number of SNPs with *p*-value < 1 × 10^−5^: number of SNPs with a suggestive *p*-value in the surrounding region.

Phenotype	Nearest Gene (Protein Coding)	Chr	Start Position	End Position	Top SNP	*p*-Value	Number of SNPs with *p*-Value < 1 × 10^−5^
NS3 (Extravagance)	*MAGI2*	7	77646393	79082890	rs13242288	9.14 × 10^−8^	24
NS4 (Disorderliness)	*CALCB*	11	14926543	15103888	rs10832317	4.56 × 10^−6^	81
HA4 (Fatigability)	*BTBD3*	20	11871371	11907257	rs4814036	2.15 × 10^−8^	72
RD (Reward Dependence)	*PARK2*	6	161768452	163148803	rs10455897	8.27 × 10^−9^	27

## Data Availability

A subset of the data is already available on the European Genome-phenome Archive (EGA) at the following links: BAM files https://www.ebi.ac.uk/ega/studies/EGAS00001000252 (accessed on 10 November 2021); sample list, vcf files https://www.ebi.ac.uk/ega/studies/EGAS00001001597 (accessed on 10 November 2021); https://www.ebi.ac.uk/ega/datasets/EGAD00001002729 (accessed on 10 November 2021). A vcf file including all the INGI variants (SNPs and INDELs) with information on allele frequencies in the whole dataset and each cohort has been submitted to the European Variation Archive (EVA) study accession number: PRJEB33648. The data are accessible at the following link: https://www.ebi.ac.uk/ena/data/view/PRJEB33648 (accessed on 10 November 2021).

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
