# Peer review of "Genetic Dissection of Temperament Personality Traits in Italian Isolates"

_genes, 2021, doi:10.3390/genes13010004_

Round 1
Reviewer 1 Report
This paper presents a study to focus on the genetic architecture of temperament, considering both its scales and subscales by performing a GWAS on Cloningerʼs Temperament and Character Inventory in 587 individuals belonging to different Italian genetic isolates. Authors shared interesting results, however, there are questions that limit my enthusiasm of the paper, as outlined below.
Comments:
- At the imputation step, authors only mentioned standard QC steps, but we need to include the details of QC for the SNPs data along with the total number of SNPs before QC, after QC (or before imputation), and after imputation. All these steps need to be clarified for the reader.
- Any multi dimensional scaling (MDS)?
- In addition, authors mentioned Info Score, but I couldn’t follow this metric for filtration. Please add the explanation for this variable.
- I assume authors applied GenABEL package in R not ABEL, am I right? If yes, please correct that package name. In addition, which method in GenABEL was considered? There are commonly used methods including FASTA or Grammar-Gamma to fit mixed linear models. Please add more details about the assigned methods. Not easy to follow by reader.
- Table S2 denotes the association analyses between outcomes and other variables including age, sec, etc. Authors mentioned regression model was considered, but I couldn’t find this method description at method section.
- In addition, Table S2 and S3 shared the beta coefficient estimate, but need more explanation as caption to explain these estimates (or beta-coefficients), along with other columns e.g., se beta (Standard error of beta estimates using mixed model!).
- Which method was considered in GTEx eQTL web browser to report the association analyses? Please add to the method section.
- How the threshold for the p-values were selected at method section? Maybe different thresholds but classifying the uncorrected p values into moderate, or strong group, etc. (Wellcome Trust Case Control Consortium 2007).
Author Response
This paper presents a study to focus on the genetic architecture of temperament, considering both its scales and subscales by performing a GWAS on Cloningerʼs Temperament and Character Inventory in 587 individuals belonging to different Italian genetic isolates. Authors shared interesting results, however, there are questions that limit my enthusiasm of the paper, as outlined below.
Comments:
Point 1. At the imputation step, authors only mentioned standard QC steps, but we need to include the details of QC for the SNPs data along with the total number of SNPs before QC, after QC (or before imputation), and after imputation total number of SNPs before QC, after QC (or before imputation), and after imputation:. All these steps need to be clarified for the reader.
Response 1. We thank the Reviewer for the comment. We added the required information in the Materials and Methods section (lines 119-137).
Point 2. Any multi dimensional scaling (MDS)?
Response 2. We thank the Reviewer for the comment. As reported in the manuscript, GWAS were conducted using mixed linear models and genomic kinship matrix was used as random effects to take into account relatedness. Although Principal Component (PC) analysis or MDS are widely used to detect population structure, a limitation of this strategy is that it fails to account for other types of sample structure, such as family structure or relatedness. Otherwise, mixed model is an approach for simultaneously addressing confounding due to population structure, family structure and relatedness. Moreover, studied support that mixed models without including PC covariates may provide better power than PC adjustment when analyzing data from human population isolates (Kang et al. Variance component model to account for sample structure in genome-wide association studies. Nat Genet. 2010 Apr; 42(4):348-54; Hoffman. Correcting for Population Structure and Kinship Using the Linear Mixed Model: Theory and Extensions. 2013. PlosOne; Population Structure and Cryptic Relatedness in Genetic Association Studies, Astle & Balding. 2009. Statistical Science).
Therefore, we did not include PCs in our GWAS.
Point 3. In addition, authors mentioned Info Score, but I couldn’t follow this metric for filtration. Please add the explanation for this variable.
Response 3. We added the explanation in the text (lines 132-136).
Point 4. I assume authors applied GenABEL package in R not ABEL, am I right? If yes, please correct that package name. In addition, which method in GenABEL was considered? There are commonly used methods including FASTA or Grammar-Gamma to fit mixed linear models. Please add more details about the assigned methods. Not easy to follow by reader.
Response 4. We thank the Reviewer for the comment. We used GenABEL/MixABEL packages and Grammar-Gamma method. We added more details in the Materials and Methods section (lines 139-145).
Point 5. Table S2 denotes the association analyses between outcomes and other variables including age, sec, etc. Authors mentioned regression model was considered, but I couldn’t find this method description at method section.
Response 5. We apologize for this lack of information. We added the explanation in the text in the Materials and Methods section (lines 112-116).
Point 6. In addition, Table S2 and S3 shared the beta coefficient estimate, but need more explanation as caption to explain these estimates (or beta-coefficients), along with other columns e.g., se beta (Standard error of beta estimates using mixed model!).
Response 6. We added this information in the caption of the tables.
Point 7. Which method was considered in GTEx eQTL web browser to report the association analyses? Please add to the method section.
Response 7. We added more details in the text (lines 217-225) in the Materials and Methods section.
Point 8. How the threshold for the p-values were selected at method section? Maybe different thresholds but classifying the uncorrected p values into moderate, or strong group, etc. (Wellcome Trust Case Control Consortium 2007).
Response 8. We thank the Reviewer for the comment. The most commonly accepted threshold is p < 5 × 10−8, which is based on performing a Bonferroni correction for all the independent common SNPs across the human genome. So, in our work, genome-wide significance was set to 5x10-8. In addition, an association was considered suggestive if it presented a p-value < 1x10-5. We added in the text the significant thresholds that we used (lines 149-152).

Reviewer 2 Report
- As multiple variants are tested for each personality traits, so there is chance that some variant which appears to be significant are false positive. Please report the pvalues after FDR correction.
- In the text it is hard to understand abbreviation of M & M, please change it to Materials and Methods
- It is not clear how author identified four genes. Please elaborate the result section
- What is the allele frequency of the identified variants in the normal population for example in the GnomAD database? Could you please check?
- The quality of the figures is poor. The labels in the Figure 2 are not readable. Please replace them with high-quality figures.
- The result in this study is from one data set only. Could you please validate the findings in other validation cohort?
Author Response
Point 1. As multiple variants are tested for each personality traits, so there is chance that some variant which appears to be significant are false positive. Please report the pvalues after FDR correction.
Response 1. We thank the Reviewer for the comment. We applied a Bonferroni correction, considering as significant the standard threshold of p-value < 5x10-8 and we considered suggestive the SNPs with p-value < 1x10-5. We did not apply other corrections. In particular, we did not correct for the number of analyzed scales because the scales are correlated to each other, as reported in other studies (https://doi.org/10.1016/j.psychres.2007.05.003, https://doi.org/10.1016/j.neuroimage.2018.08.038) and as detected in our population. We added in the text more information (lines 149-152). In addition, we reported the correlations between the scales in our samples (lines 111-112; 231-238 and Table S2).
Point 2. In the text it is hard to understand abbreviation of M & M, please change it to Materials and Methods.
Response 2. Sorry for the inaccuracy. We changed it (line 285).
Point 3. It is not clear how author identified four genes. Please elaborate the result section.
Response 3. We thank the Reviewer for the comment that allows us to improve the clarity of the manuscript. We added further details in the Results section (lines 257-286).
Point 4. What is the allele frequency of the identified variants in the normal population for example in the GnomAD database? Could you please check?
Response 4. We thank the Reviewer for the comment. We added this information in the VEP annotation (see Table S5) both for 1000 Genomes and GnomAD databases. In particular it is important to underline that the individuals involved in the study are not patients but voluntary participants recruited within the "Friuli Venezia Giulia Genetic Park" project.
Point 5. The quality of the figures is poor. The labels in the Figure 2 are not readable. Please replace them with high-quality figures.
Response 5. We replaced the figures following the parameters indicated by the journal.
Point 6. The result in this study is from one data set only. Could you please validate the findings in other validation cohort?
Response 6. We are aware that the data reported in this work are from one population, but unfortunately, to our knowledge, there are no other cohorts with genotypes and phenotypes data comparable with ours and ready to make a quick replication. Following this work, we will surely look for other groups working with this kind of phenotypes in both inbred and outbred populations in order to provide a replication of our most interesting findings. We added a sentence at the end of the manuscript (line 516) to underline this lack.

Round 2
Reviewer 1 Report
All the comments have been addressed. Thank you!
Author Response
Point 1: All the comments have been addressed. Thank you!
Response 1: We thank the Reviewer again for the comments.